# Factors associated with Facebook addiction among university students amid the COVID-19 pandemic: Findings from an online cross-sectional survey

Rezaul Karim Ripon[1]*, Abdullah Al Zubayer[2], Quazi Maksudur Rahman[1], Abid Hasan Khan[1], Arifur Rahaman[3], M. Tasdik Hasan[4,5], Md. Rifat Al Mazid Bhuiyan[6], Md. Kamrul Ahsan Khan[7], Md. Ashraf Uddin Chowdhury[7], Md. Zakir Hossain[7]

1 Department of Public Health and Informatics, Jahangirnagar University, Savar, Dhaka, Bangladesh, 2 Department of Sociology, University of Barishal, Barishal, Bangladesh, 3 Department of Sociology, University of Dhaka, Dhaka, Bangladesh, 4 Jeeon Bangladesh Ltd., Dhaka, Bangladesh, 5 Public Health Foundation, Bangladesh, Dhaka, Bangladesh, 6 Dhaka Community Medical College, Mogbazar, Dhaka, Bangladesh, 7 Sheikh Sayera Khatun Medical College, Gopalgonj, Bangladesh

* riponrezaul5@gmail.com

## Abstract

### Background

Facebook addiction (FA) has been suggested as a potential behavioral addiction. There is a severe lack of research evidence regarding the Facebook addiction behavior among university students during the ongoing COVID-19 pandemic. The aim of this study was to determine factors associated with Facebook addiction among Bangladeshi university students.

### Methods

A cross-sectional online survey was conducted among 2,161 Bangladeshi university students during the COVID-19 pandemic from June 2021 to September 2021. A well fitted regression model in R programming language was used for this study.

### Results

Female respondents and those whose family monthly income was <25,000 BDT were more addicted to Facebook than other respondents. Respondents who lost a family member or a relative to COVID-19, engaged in physical activities (exercise) during the pandemic, used Facebook for work purposes or used Facebook to relieve daily stress were more addicted to Facebook.

### Conclusion

Overuse of social media is problematic as it can trigger several mental health symptoms, especially among students. Adequate and effective interventions are required to educate students about the dangers of Facebook addiction and to provide an alternative, healthy options.

**Data Availability Statement:** All relevant data are within the manuscript and its Supporting Information files.

**Funding:** there has been no significant financial support for this work that could have influenced its outcome.

**Competing interests:** no conflicts of interest associated with this publication.

**Abbreviations:** BDT, Bangladesh taka; FA, Facebook Addiction.

## Introduction

Our lives have been simplified by the growth of technology-mediated help in delivering better communication services, but behavioral addictions such as internet addiction and social media addiction have become common as a result [1, 2]. Facebook has grown in popularity among many social networking sites. Facebook currently has 2.895 billion monthly active users and almost 125.46 million users in Bangladesh, among them 43 million user ages were 19–24 years old [3]. Young people tend to be the primary users of Facebook and other social networking sites, and for them, excessive usage of Facebook might be addictive [4, 5]. New hand-held gadgets, such as smartphones, tablets, and laptop computers, are increasing Internet access and portability, making it feasible to work online and enjoy leisure activities while on the go. Facebook addiction (FA) can be classified as one of the aspects of Internet addiction (IA) [6]. Facebook addiction is a term coined by researchers that are applied to individuals who engage in excessive, compulsive Facebook use for the purposes of mood alteration, with negative personal outcomes [7]. Facebook addiction refers to dependency on Facebook caused by excessive use, which disrupts daily activities [8]. Problematic Facebook use has been defined as Facebook use that creates problems in users' lives, such as psychological, emotional, social, school, or work difficulties [9]. The six criteria of addiction (salience, mood modification, tolerance, withdrawal, conflict, and relapse) or similar factors based on the definition of gambling addiction (e.g., withdrawal, interpersonal problems due to Facebook use, time management, and performance problems) are used to define Facebook addiction [10, 11]. Reasons and motivations behind Facebook's popularity include being able to easily access updated features which facilitate communicating with friends, meeting others based on shared interests, sharing pictures and videos, blogging, dating, and even gaming [1, 5, 12], on the other hand, stated that people's activities on Facebook might include things like gaming and gambling [13], in addition to social networking. Most Facebook users find comfort, life satisfaction, happiness, and social support while using this platform, although, others are connected with it as a means of escapism [7, 14]. Facebook addiction is associated with many negative psychological traits like self-inferiority and depression [15]. Ordinary Facebook users differ statistically in terms of self-esteem and life satisfaction from both addicted and intensive users, as addictive and intensive users are found to be low self-esteemed and less satisfied with life [16].

Numerous factors have been reported in relation to social media use more generally, and Facebook use more specifically. Sociodemographic factors such as age, gender, relationship status, and occupational/educational status can all play important roles in determining patterns of Facebook use [17]. Predictors of problematic Facebook use include a wide range of activities and factors including less engagement in physical activity [14, 18] loneliness [2, 19], poor sleep [20], and relationship disengagement [21]. Extraversion, narcissism, high degrees of neuroticism, and low levels of self-esteem all have a strong correlation with excessive Facebook use [22]. Shyness, loneliness, and negative affect were found to be positively connected with Facebook Addiction [23]. Besides, failure in love, a history of domestic violence, a stressful life event, and sleep disturbances were all identified as risk factors for Facebook addiction in a previous study in Bangladesh [17]. SARS-CoV-2, or COVID-19, has emerged as the most serious public health threat of the contemporary age. No other events in the last few decades have had half the devastating effect as the COVID-19 pandemic is having now. More than 5 million individuals have died as a result of this infectious virus, which has infected about 200 million people. The coronavirus spread quickly across the world due to its infectious nature, resulting in a large number of deaths. COVID-19 brought about many changes in daily lives and lifestyles around the globe (e.g., working from home, home quarantine, home-schooling, etc.) and many new norms as preventive measures (e.g., wearing masks, sanitation, social

distancing, and vaccination). All these together formed a 'new normal' environment coping with the challenges of the COVID-19 pandemic while conforming to preventive measures [24]. In many ways, COVID-19 had an impact on nearly every nation in economic, social, cultural, political, and other spheres. Addiction to social media sites increased during COVID- 19 owing to quarantine, social distancing, and other health measures [25]. Several studies have been conducted on smartphones and Internet addiction during the first wave of the COVID-19 pandemic [26, 27]. The objectives of this paper were is to determine the factors associated with Facebook Addiction among university students, as well as make possible significant recommendations for taking appropriate preventive measures during the future waves of COVID-19, while undergoing a long-term homestay unlike any other. To assess the Facebook addiction behavior among the students, the study used the Bergen Facebook Addiction Scale (BFAS). The Bergen scale has been validated by several studies in Bangladesh [6, 8].

## Methods

### Participants and procedure of data collection

During the COVID-19 pandemic in mid- 2021, this online cross-sectional survey was conducted to assess addiction to Facebook and its associated factors among university students. The survey took place from June 2021 to September 2021. Participants were recruited through social media as per convenience sampling by using Google Form. Due to the COVID-19 pandemic, there were not possible to collect data through a face-to-face interview. We collected data by google form. The Google forms had several advantages which a manual questionnaire does not have, namely paperless, environmentally friendly, time-efficient, labour costs, accurate recapitulation of respondents' answers, and practical. Inclusion criteria included: being a university student (age at least 18 years or older), having internet access, and residing in Bangladesh during the study. Those who are below 18 years old, not willing to participate, and missing data were excluded. A sufficient number of research assistants were recruited to get a high response rate in the survey. On the first page of the electronic survey, the participants gave their written consent by using the yes/ no option. The survey took approximately 15/20 minutes to complete. Participation in this survey was on a volunteer basis and was anonymous. The study was ethically approved by Sheikh Sayera Khatun Medical College, Gopalgonj ethical committee (SSKMC/EC/2021/476(B)). Initially, there were 2,170 responses, and after removing incomplete data, there were a total of 2161 responses whose ages were 18 or above years old and willing to participate in the study.

### Measures

The aim of this study was to identify the prevalence and factors associated with Facebook addiction among university students amid the COVID-19 pandemic. The structured Google form had several sections: a) socio-demographic information alongside a participant consent form. b) COVID-19-related information; and c) the Burgen Facebook Addiction Scale (BFAC) [10].

**Socio-demographic measures.** Information on socio-demographic variables including age (= < 20 years, 21 to 25 years, 26 to 30 years, 30 years or more), gender (Female, Male), educational background (Science, Business studies, social science, Arts, Humanitarian studies, others), family monthly income (<25,000 BDT, 25,000–50,000 BDT, >50,000 BDT), marital status (bachelor, married, others), type of family (nuclear family, combined Family), current location (rural, urban).

**Addiction to Facebook.** A validated Facebook addiction scale (Burgen Facebook Addiction BFAC) [10], is a six-item self-reported scale that is a brief and effective psychometric

instrument for assessing at-risk social media addiction on the internet, used to assess addiction on Facebook. It is a 5-point Likert scale with (1) very rarely, (2) rarely, (3) sometimes, (4) often, and (5) very often. The overall score ranged from 6 to 30, with higher scores reflecting a greater addiction to Facebook [10]. Using a score of 3 or more in response to four of the six items is an indicator of addiction [10].

## COVID-1 9 related information

COVID-19-related information consists of twelve questions. We run a pilot study for these 12 questions before rolling it out for this study. Before the initiation of data collection, we conducted a pilot study to determine whether the study's questionnaires were understandable for the general public, particularly for university students, and to examine the viability of the data collection tools for gathering data swiftly without imposing onerous conditions. Upon completing the pilot study, we revised some questions and altered wordings as per the convenience of the respondents. In addition, the pilot study was also conducted to determine whether the data showed too much or too little variability and to cross-check the eligibility criteria for the respondents. To assure validity, the researchers committed to upholding the study's credibility at every stage of the data collection and analysis process. We further reinforced the credibility by ensuring that each respondent understood the questionnaire and the purpose of the study. Both reliability and sensitivity were good. The Cronbach alpha for these questions was 0.864. The question responses were yes or no. *Are you separated from your family because of COVID-19*? *Were you in quarantine during this pandemic*? *Do you feel detached from your friends/peers in this pandemic*? *Are you dealing with relationship issues like family feuds or break ups*? *Do you feel lonely*? *Is your movement restricted because of a lockdown*? *Do you feel depressed about this pandemic*? *Have you lost any family members or relatives to COVID-19*? *and have you thought about suicide at least once in this pandemic situation*? *and are you involved in any physical activities (exercise)*? *How much time (hours) on average do you spend on Facebook in a day*? *and What is your reason for using Facebook*? *(a. Work; b. Education; c. Upkeep of social networks. d. Relief from day-to-day stress e. Passage of time f. Absence of a specific reason).*

## Statistical analysis

Descriptive statistics (frequencies, percentages) were computed for the prevalence. A crosstab was conducted for Facebook addiction and COVID-19-related questions, and for sociodemographic variables. A well-fitted regression model was used to assess the association between Facebook addiction and COVID-19 relationship-related questions. All analyses were carried out with a p-value less than 0.05 using the R programming language.

# Results

## Demographic analysis

A total 2,161 respondents were included in this study. Among them Age: = < 20 years (265, 12.3%), 21 to 25 years(1724, 79.8%), 26 to 30 years (146,6.8%), 30 years or more(26,1.2%); Gender:Female(1130,52.3%), Male(1031,47.7%); Educational background: Science (1142,52.8%), Business studies (307,14.2%), Social science(459,21.2%), Arts (134,6.2%), Humanitarian studies (119,5.5%), others(0%); Family Monthly Income: <25,000 BDT (783,36.2%), 25,000–50,000 BDT(904,41.8%), >50,000 BDT (474,21.9%); Marital status: Bachelor(1930,89.3%), Married (224,10.4%), Others (7,.3%); Type of family: Nuclear family(1821, 84.3%), Combined Family (340,15.7%); Current location: Rural (.03,27.9%), Urban (1558,72.1%) [Table 1].

## Association of the demographic variable and responses with addiction in Facebook

The overall Facebook addiction among university students in Bangladesh was 28.3% (n = 612). Among the respondent of bachelor marital status were 89.31%(n = 548), Nuclear family 84.27% (n = 509), Urban responded 72.10% (n = 438), Age 21 to 25 years 79.78% (n = 492), Female responded 52.29% (n = 348), Educational background Science 52.85%(n = 302), Family Monthly Income 25,000–50,000 BDT 41.83% (n = 254). Female respondents were 1.3 times more addicted to Facebook than male [OR (95% of CI, p value):1.293(1.071–1.561), .008]. Family Monthly Income <25,000 BDT were1.4 times more addicted to Facebook than those monthly income were >50,000 BDT [OR (95% of CI, p value): 1.430(1.103–1.853), .007]. And they remain significant after adjusting all other variables [Table 2, Fig 1].

## Association of the COVID-19 related question and positive response with addiction on Facebook

Because of COVID-19 3.24%(n = 70) separated from their family, 13.84%(n = 299) were in quarantine during this pandemic, 19.81%(n = 428) feel detached from their friends/peers in this pandemic, 16.66%(n = 360) were dealing with relationship problem such as family conflicts or break-ups in this pandemic. 19.11%(n = 413) feel lonely in this pandemic. 24.90% (n = 538) movement were restricted because of lockdown, 24.20%(n = 523) feel depressed in this pandemic, 7.08%(n = 153) lost any family member or relatives to COVID-19, 9.76% (n = 211) thought about suicide at least once in this pandemic situation, 8.98%(n = 194)

**Table 1. Prevalence demographic information.**

| Demographic Variables | | Frequency (%) |
|---|---|---|
| **Age** | = < 20 years | 265(12.3%) |
| | 21 to 25 years | 1724(79.8%) |
| | 26 to 30 years | 146(6.8%) |
| | 30 years or more | 26(1.2%) |
| **Gender** | Female | 1130(52.3%) |
| | Male | 1031(47.7%) |
| **Educational background** | Science | 1142(52.8%) |
| | Business studies | 307(14.2%) |
| | Social science | 459(21.2%) |
| | Arts | 134(6.2%) |
| | Humanitarian studies | 119(5.5%) |
| **Family Monthly Income** | <25,000 BDT | 783(36.2%) |
| | 25,000–50,000 BDT | 904(41.8%) |
| | >50,000 BDT | 474(21.9%) |
| **Marital status** | Bachelor | 1930(89.3%) |
| | Married | 224(10.4%) |
| | Others | 7(.3%) |
| **Type of family** | Nuclear family | 1821(84.3%) |
| | Combined Family | 340(15.7%) |
| **Current location** | Rural | 603(27.9%) |
| | Urban | 1558(72.1%) |
| **Addiction in Facebook** | Yes | 612(28.3%) |
| | No | 1549(71.7%) |

**Table 2. Association of the demographic variable and respond with addicted in Facebook.**

| Demographic Variable | | Addicted in Facebook | | |
|---|---|---|---|---|
| | | Frequency (%) | OR (95% C.I.), P value | aOR (95% C.I.), P value |
| Age | = < 20 years | 81(12.26%) | .832(.356–1.944), .670 | .727(.297–1.777), .484 |
| | 21 to 25 years | 492(79.78%) | .754(.334–1.704), .498 | .677(.287–1.596), .372 |
| | 26 to 30 years | 30(6.76%) | .489(.198–1.204), .120 | .477(.189–1.200), .116 |
| | 30 years or more | 9(1.20%) | 1 | 1 |
| Gender | Female | 348(52.29%) | 1.293(1.071–1.561), .008 | 1.323(1.087–1.610), .005 |
| | Male | 264(47.71%) | 1 | 1 |
| Educational background | Science | 302(52.85%) | .766(.510–1.152), .200 | .833(.548–1.265), .391 |
| | Business studies | 85(14.21%) | .816(.516–1.292), .386 | .907(.568–1.447), .682 |
| | Social science | 147(21.24%) | 1.004(.652–1.548), .984 | 1.078(.693–1.677), .738 |
| | Arts | 40(6.20%) | .907(.532–1.548), .720 | .962(.560–1.650), .887 |
| | Humanitarian studies | 38(5.51%) | 1 | 1 |
| Family Monthly Income | <25,000 BDT | 244(36.23%) | 1.430(1.103–1.853), .007 | 1.455(1.107–1.913), .007 |
| | 25,000–50,000 BDT | 254(41.83%) | 1.234(.956–1.594), .107 | 1.260(.972–1.634), .081 |
| | >50,000 BDT | 114(21.93%) | 1 | 1 |
| Marital status | Bachelor | 548(89.31%) | 1.036(.761–1.411), .822 | .873(.162–4.699), .874 |
| | Married | 62(10.37%) | .775 | .780(.143–4.265), .775 |
| | Others | 2(0.32%) | 1 | 1 |
| Type of family | Nuclear family | 509(84.27%) | .893(.693–1.150), .379 | .895(.656–1.221), .482 |
| | Combined Family | 103(15.73%) | .873(.162–4.699), .874 | 1 |
| Current location | Rural | 174(27.90%) | 1.037(.842–1.277), .731 | .976(.779–1.222), .831 |
| | Urban | 438(72.10%) | 1 | 1 |

involved in any physical activities (exercise), during pandemic on average you spend on Facebook in a day: 0 to 5 hours 0.74%(n = 232), 6 to 10 hours 10.18%(n = 220), 11 hours or more 7.40%(n = 160); reason for using Facebook: Work 1.48%(n = 32), Maintain social Network

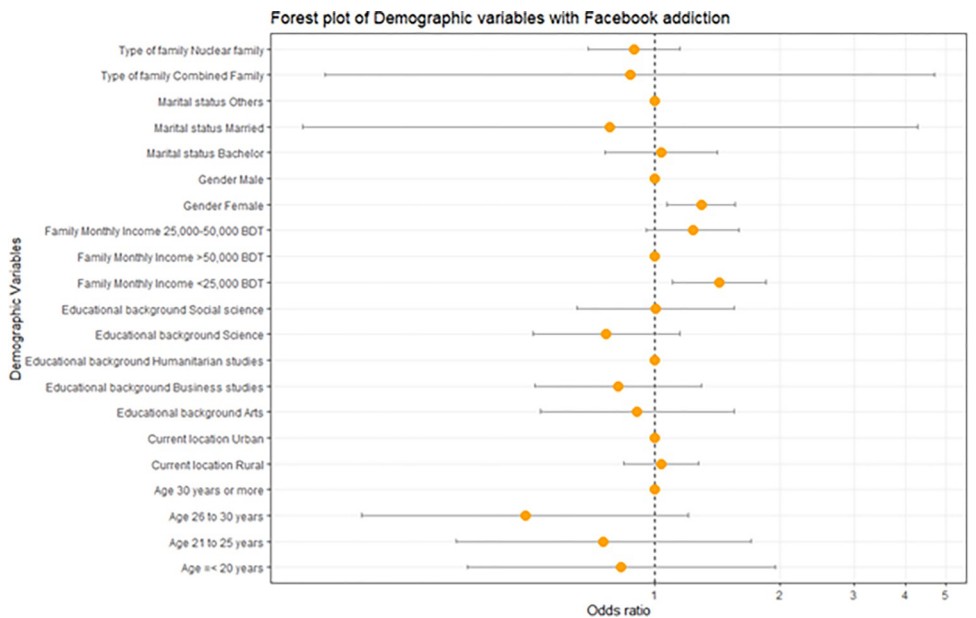

**Fig 1. Forest plot of demographic variables with Facebook addiction.**

Table 3. Association of the COVID-19 related question and positive respond with addicted in Facebook.

| COVID-19 related questions * | | Addicted in Facebook | | |
|---|---|---|---|---|
| | | Frequency (%) | OR(95% CI), p value | aOR(95% CI), p value |
| Are you separated from your family because COVID-19? | | 70(3.24%) | 948(.705–1.275), .725 | 1.139(.821–1.580), .437 |
| Were you in quarantine in this pandemic? | | 299(13.84%) | .958(.794–1.155), .650 | 1.214(.984–1.498), .071 |
| Do you feel detached from your friends/peers in this pandemic? | | 428(19.81%) | 2.530(2.072–3.088), .0001 | .645(.508–.819), .0001 |
| Are you dealing with relationship problem such as family conflicts or break ups? | | 360(16.66%) | .435(.360-.527), .0001 | .712(.570-.889), .003 |
| Do you feel lonely? | | 413(19.11%) | .322(.264-.392), .0001 | .604(.473-.770), .0001 |
| Is your movement restricted because of lockdown? | | 538(24.90%) | .623(.473-.821), .001 | .969(.702–1.337), 848 |
| Do you feel depressed in this pandemic? | | 523(24.20%) | .374(.292–.480), .0001 | .647(.484-.864) |
| Have you lost any family member or relatives to COVID-19? | | 153(7.08%) | 1.322(1.059–1.649), .013 | .863(.675–1.103), .238 |
| Have your thought about suicide at least once in this pandemic situation? | | 211(9.76%) | .412(.334-.509), .0001 | .809(.633–1.035), .091 |
| Are you involved in any physical activities (exercise)? | | 194(8.98%) | 1.497(1.228–1.825), .0001 | 1.2439(.998–1.547), .052 |
| How much time (hours) on average you spend on Facebook in a day? | 0 to 5 hours | 232(10.74%) | .132(.112-.984), .005 | .282(.212-.374), .0001 |
| | 6 to 10 hours | 220(10.18%) | .214(.101-.954), .004 | .647(.481-.872), .004 |
| | 11 hours or more | 160(7.40%) | 1 | 1 |
| What is your reason for using Facebook? | Work | 32(1.48%) | 1.19(1.24–3.1), .042 | 1.619(1.044–2.5090), .031 |
| | Maintain social_Network | 171(7.91%) | 1.8(.765–2.05), .087 | 1.258(.985–1.605), .066 |
| | Relief_ daily stress | 222(10.27%) | 1.536(1.214–2.246), .003 | 1.376(1.084–1.746), .009 |
| | Time_ passing | 232(10.74%) | .641(.123-.740), .06 | .741(.583-.940), .014 |
| | No-reason | 390(18.05%) | .12(.10-.653), 0.32 | .692(.555-.863), .05 |

• No used as a reference category.

7.91%(n = 171), Relief daily stress 10.27%(n = 222), Time passing 10.74%(n = 232), No-reason 18.05%(n = 390). Those feeling detached from your friends/peers in this pandemic were 2.5 times more addicted to Facebook than those who are not [2.530(2.072–3.088), .0001], this variable also remains significant after adjusting all other variables. Respondents who lost any family member or relatives to COVID-19 were 1.3 times higher addicted to Facebook than those are not [OR (95% of CI), p value: 1.322(1.059–1.649), .013] and this variable didn't significant after adjusting all other variables. And those involved in any physical activities (exercise) in this pandemic were 1.5 times more addicted to Facebook than those are not [OR (95% of CI), p value: 1.497(1.228–1.825), .0001] and this variable also didn't significant after adjusting all other variables. Those who responded to using Facebook for Work purposes were1.2 times more addicted to Facebook than those are not [OR (95% of CI), p value: 1.19(1.24–3.1), .042], responded use Facebook for Relief of daily stress purposes were 1.5 times more addicted in Facebook than those are not [OR (95% of CI), p value: 1.536(1.214–2.246), .003] and these variables remain significant after adjusting all other variables [Table 3].

## Discussion

This study explored several factors associated with Facebook addiction among university students. Our study result shows that females were more addicted to Facebook than their male counterparts. Prior studies found that the COVID-19 outbreak has severely constrained people's daily lives and increased their use of social media [14] and about forty percent of women reported problematic social media (Facebook) use [28]. Besides it was seen that who's monthly

family income was less than twenty-five thousand (BDT) were greater addicted to Facebook than others. Previous studies disclosed that the level of income does not show a significant influence on the risk of Facebook addiction [29]. The difference in results based on income appears may be due to geographical location and time variation (pandemic situation). Our study found that students who felt disconnected from friends or peers during the COVID-19 pandemic had higher Facebook addiction, whereas a previous study found that physical separation encouraged people to use the Internet for virtually all daily tasks during the pandemic [30]. In our study, it was also highlighted that students who lost any family member or relative to COVID-19 were more addicted to Facebook, which may be the reason for suffering from a stressful life event due to the loss of beloved members. This is similar to prior studies in Bangladesh that found having a stressful life event was responsible for Facebook addiction [17]. This study revealed that those involved in any physical activities in this pandemic had higher Facebook addiction. A prior study also showed that problematic smartphone use is linked to physical inactivity during this COVID-19 outbreak [31] and that these trends in smartphone use may increase [31]/decrease [32] the use of Facebook, which turns into an addiction. Other studies also revealed that the increase in physical activities reduces/increase Facebook addiction [14, 32, 33]. As an adaptation measure to personal and work life, COVID-19-induced lockdown resulted in people becoming more attached to their smartphones [34, 35], and such problematic use may lead them to use their smartphones and social media sites even during any type of physical activity like a workout. But the inconsistency between these two findings varies due to unclear reasons. This inconsistency may occur for the "Internet Plus Exercise" campaign to promote health.

However, our study reported that students using Facebook for work purposes (e.g., online marketing) were more addicted to Facebook, whereas a previous study showed that broadcasting behavior on Facebook positively predicts Facebook addiction [36]. Furthermore, this study discovered that students who used Facebook to relieve daily stress were more likely to become Facebook addicts. Previous studies considered that time spent using Facebook is one of the main predictors of Facebook addiction [37], and daily stress was linked to the amount of time spent on Facebook and the likelihood of becoming addicted to it. The relationship between daily stress and Facebook use intensity was negatively moderated by perceived offline social support, implying that people who got low levels of support offline were more inclined to increase their Facebook use when they were under stress [38]. Our study has highlighted some of the key factors influencing Facebook addiction among university students after one year of the COVID-19 pandemic. Along with many other factors of Facebook addiction, the COVID-19 pandemic has resulted in some factors such as low income, disconnection and loss of family members. Governmental and non-governmental organizations should come forward to take appropriate strategies to overcome this vulnerable condition, such as Facebook addiction, for the well-being of university students to protect the future leaders of the nation.

## Limitations of the study

There were a few drawbacks to this study. To begin with, the study's self-reported data was prone to reporting bias, and respondents from lower socioeconomic classes who do not have access to the internet or Wi-Fi were not included in the study. Second, selection bias was a limitation of the convenience sampling technique. Finally, due to the study's cross-sectional design, it wasn't easy to investigate any possible causality. Further research should be done using a mixed-methods approach with large-scale studies. Despite the study's limitations, we felt it provides vital evidence on university students' Facebook addiction after a year of living with the COVID-19 outbreak.

## Supporting information

**S1 Dataset.**
(XLSX)

## Acknowledgments

The authors would like to express their heartiest gratitude to all the volunteers in our team for their voluntary contributions during the data collection period by sharing the survey link on various online platforms: Umme Laiba Raha (Geography and Environmental Studies, University of Chittagong), Labanya Bhowmik (Agricultural Economics and Rural Sociology, Bangladesh Agricultural University), Nur E Nasren Joty (BBA, University of Liberal Arts Bangladesh), Mahmudul Hasan (Sociology, Jagannath University), Anika Mahzabin Sandhi (Sociology and Anthropology, Green University of Bangladesh), Mehnaz Rahman Shithy (Criminology And Police Science, Mawlana Bhashani Science And Technology), Sadia Zaman Akhtar (Health Economics, University of Dhaka), Saba Tabassum (Disaster and Human Security Management, Bangladesh University of Professionals), Anas Al Masud (Public Administration, Bangladesh University of Professionals), Sadia Farzana (Biochemistry and Molecular Biology, Bangabandhu Sheikh Mujibur Rahman Science and Technology University), Alfi Shahrin (Pharmacy, East West University), Israt Jahan Akhi (International Relations, Jahangirnagar University), Marzia Feruz Snigdha (Public Health, State University of Bangladesh), Israt Jahan Prapti (Sociology, Noakhali science and Technology University), Safayat Sikder Sakib (Law, East West University), Tanzila Aktar Tisha (Pharmacy, East West University), Mina Ferdious (Mass Communication and Journalism, Bangladesh University of professionals), Sumaiya Zannat (Medicine, Armed Forces Medical College), Samsun Nahar Priya (Biochemistry and Molecular Biology, Mawlana Bhashani Science and Technology University), Israt Zahan (Economics, University of Dhaka), Md Junayeth Bhuiyan (Population Science, Jatiya Kabi Kazi Nazrul Isalm University), Mimi Akter (IER, Chittagong University), Pronab Kumar Paul Partha (Agricultural Engineering, Bangladesh Agricultural University), Md. Nayeem Hasan Pramanik (Economics, Pabna University of Science and Technology), Md Delower Hossain (Pharmacy, State University of Bangladesh), Tabara Rahman (Economics, Jagannath University), Anonna Das Srishty (Sociology, Shahjalal University of Science and Technology), Arifa Akter Tangina (Economics, Comilla University).

### Declaration

Ethics approval and consent to participate: SSKMC/EC/2021/476(B).

## Author Contributions

**Conceptualization:** Quazi Maksudur Rahman, Abid Hasan Khan, Arifur Rahaman.

**Data curation:** Rezaul Karim Ripon.

**Formal analysis:** Rezaul Karim Ripon.

**Investigation:** Abdullah Al Zubayer.

**Methodology:** Rezaul Karim Ripon.

**Project administration:** Abdullah Al Zubayer, Quazi Maksudur Rahman, Abid Hasan Khan, Md. Rifat Al Mazid Bhuiyan, Md. Kamrul Ahsan Khan, Md. Ashraf Uddin Chowdhury, Md. Zakir Hossain.

**Resources:** Md. Rifat Al Mazid Bhuiyan, Md. Kamrul Ahsan Khan, Md. Ashraf Uddin Chowdhury, Md. Zakir Hossain.

**Software:** Rezaul Karim Ripon.

**Writing – original draft:** Rezaul Karim Ripon, Quazi Maksudur Rahman, Arifur Rahaman.

**Writing – review & editing:** Rezaul Karim Ripon, M. Tasdik Hasan.

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
