## [Decision Letter · Decision Letter 0]

7 Jun 2022

PONE-D-22-08770Factors Associated with Facebook Addiction among Bangladeshi University Students amid the COVID-19 Pandemic: Findings from an Online Cross-Sectional SurveyPLOS ONE

Dear Dr. Ripon,

Thank you for submitting your manuscript to PLOS ONE. After careful consideration, we feel that it has merit but does not fully meet PLOS ONE’s publication criteria as it currently stands. Therefore, we invite you to submit a revised version of the manuscript that addresses the points raised during the review process.

We look forward to receiving your revised manuscript.

Kind regards,

Latika Gupta

Academic Editor

PLOS ONE

Journal Requirements:

“no”

“NO”

Additional Editor Comments:

Please consider enriching methods further with standard guidelines for reporting survey based studies e.g. CHERRIES checklist, 10.3346/jkms.2020.35.e398.

Please discuss limitations inherent to a survey based study.

Please address reviewer comments appended below.

Reviewers' comments:

Reviewer's Responses to Questions

**Comments to the Author**

1. Is the manuscript technically sound, and do the data support the conclusions?

Reviewer #1: Yes

Reviewer #2: Yes

2. Has the statistical analysis been performed appropriately and rigorously? 

Reviewer #1: Yes

Reviewer #2: Yes

3. Have the authors made all data underlying the findings in their manuscript fully available?

Reviewer #1: Yes

Reviewer #2: Yes

4. Is the manuscript presented in an intelligible fashion and written in standard English?

Reviewer #1: No

Reviewer #2: Yes

5. Review Comments to the Author

Reviewer #1: The research idea and manuscript is indeed interesting and relevant in today’s world

The manuscript would benefit grammatical revisions

If abbreviations are being used, they should be use consistently throughout the paper.

Abstract:

In the background- There could be more generalizability to the topic. Rather than using the phrase Bangladeshi university students, is this study generalizable to all university students during the covid-19 pandemic

Methods

What is Google form?

COVID-19 related information questionnaire- what is the origin of these questions? Are they derived from any validated survey or the authors self-validated it by running a pilot?

How is this information different from other studies performed about Facebook addiction? Your results are quite similar to studies that were performed in pre covid-19 pandemic era. So, how would you factor in the COVID-19 pandemic and its resultant isolation in your study.

The result that you have reported that physical activity is associated with increased Facebook addiction is different from previous studies. But no thought process is there to try to explain the difference. Could it be, for example that while being on an exercise machine people use social media to divert attention?

I have made comments to highlighted areas in a pdf copy of the manuscript as well

Reviewer #2: Nice manuscript. The authors wants to determine the factors associated with FA among

university students.

English writing needs some improvement.

Discuss only the results/ the concept should be in the introduction section.

Following the author's recommandations (table and results). Results: all Tables need some editing.

References: typo in accord with the guidelines authors

6. PLOS authors have the option to publish the peer review history of their article (what does this mean?). If published, this will include your full peer review and any attached files.

Reviewer #1: **Yes: **Tulika Chatterjee

Reviewer #2: No

---

## [Author Response · Author response to Decision Letter 0]

20 Jun 2022

Reviewer #1: The research idea and manuscript is indeed interesting and relevant in today’s world

Author Responses: Thanks for your complement for our working.

The manuscript would benefit grammatical revisions

Author Responses: Thanks for your comment. We revised it.

If abbreviations are being used, they should be use consistently throughout the paper.

Author Responses: Thanks for your comment. We revised it in line 255.

Abstract:

In the background- There could be more generalizability to the topic. Rather than using the phrase Bangladeshi university students, is this study generalizable to all university students during the covid-19 pandemic

Author Responses: Thanks for your comment. We revised it in line 4-24.

Methods

What is Google form?

Author Responses: Thanks for your comment. Due to the COVID-19 pandemic, there were not possible to collect data through a face-to-face interview. We collected data by google form. The Google forms had several advantages which a manual questionnaire does not have, namely paperless, environmentally friendly, time-efficient, labour costs, accurate recapitulation of respondents' answers, and practical. Line 96-100.

COVID-19 related information questionnaire- what is the origin of these questions? Are they derived from any validated survey or the authors self-validated it by running a pilot?

Author Responses: Thanks for your comment. COVID-19-related information consists of twelve questions. We run a pilot study for these 12 questions before rolling it out for this study. Both reliability and sensitivity were good. The Cronbach alpha for these questions was 0. 864. Line 132-134.

How is this information different from other studies performed about Facebook addiction?

Author Responses: Thanks for your comment. 

In the first half of the COVID-19 pandemic, there were some studies regarding Facebook addiction in Bangladesh, but there is no study who find out the association of Facebook addiction and COVID-19 related information in Bangladesh. Many studies suggested that Facebook addiction is increased for COVID-19 related information(https://doi.org/10.3389/fpsyg.2021.682837). For that, we included these questions. This was one of the unique for our study. Also, this study will help to trend of Facebook addiction in Bangladesh. 

Your results are quite similar to studies that were performed in pre covid-19 pandemic era. So, how would you factor in the COVID-19 pandemic and its resultant isolation in your study.

Author Responses: Thanks for your comment. 

There were a few studies conducted in Bangladeshi University in pre pandemic. To conduct the actual scenery more research had to conduct. Although two studies in pre pandemic in Jahangirnagar University shown Facebook addiction can be addiction was 39.7%-78.8% and both studies were completed in 2018 with 300 sample sizes. (https://www.neliti.com/publications/263131/impact-of-facebook-obsession-among-university-students-in-bangladesh, DOI: 10.1016/j.psychres.2018.12.039, https://doi.org/10.1016/j.jval.2018.04.1284). That mean there were huge variation of the percentages. Many research showed peoples fear and psychological subject were prone to decline trend after the 1st wave of Covid-19( http://dx.doi.org/10.1136/bmjopen-2020-040620, https://doi.org/10.2147/PRBM.S354083) . Many people are starting to ignoring the national guideline (https://www.aa.com.tr/en/asia-pacific/2nd-wave-of-covid-19-hits-bangladesh/1983091) of COVID-19 after 1st wave. Our study conducted in the mid-2021 of COVID-19 pandemic in Bangladesh. Although COVID-19 information is linked to Facebook addiction(https://doi.org/10.3389/fpsyg.2021.682837). That can affect our result. 

The result that you have reported that physical activity is associated with increased Facebook addiction is different from previous studies. But no thought process is there to try to explain the difference. Could it be, for example that while being on an exercise machine people use social media to divert attention?

Author Responses: Thanks for your comment. 

This study revealed that those involved in any physical activities in this pandemic had higher Facebook addiction. A prior study also showed that problematic smartphone use is linked to physical inactivity during this COVID-19 outbreak [37] and that these trends in smartphone use may increase [37]/decrease [38] the use of Facebook, which turns into an addiction. Other studies also revealed that the increase in physical activities reduces/increase Facebook addiction. [14] [31] [38]. As an adaptation measure to personal and work life, COVID-19-induced lockdown resulted in people becoming more attached to their smartphones [32] [33], and such problematic use may lead them to use their smartphones and social media sites even during any type of physical activity like a workout. But the inconsistency between these two findings varies due to unclear reasons. This inconsistency may occur for the “Internet Plus Exercise” campaign to promote health.

I have made comments to highlighted areas in a pdf copy of the manuscript as well

Reviewer #2: Nice manuscript. The authors want to determine the factors associated with FA among university students.

Author Responses: Thanks for your compliment. 

English writing needs some improvement.

Author Responses: Thanks for your comment. We revised it.

Discuss only the results/ the concept should be in the introduction section.

Author Responses: Thanks for your comment. We revised it.

Following the author's recommendations (table and results). Results: all Tables need some editing.

Author Responses: Thanks for your comment. We revised it.

References: typo in accord with the guideline’s authors

Author Responses: Thanks for your comment. We revised it.

---

## [Editor Report · Decision Letter 1]

19 Jul 2022

PONE-D-22-08770R1Factors Associated with Facebook Addiction among University Students amid the COVID-19 Pandemic: Findings from an Online Cross-Sectional SurveyPLOS ONE

Dear Dr. Ripon,

Thank you for submitting your manuscript to PLOS ONE. After careful consideration, we feel that it has merit but does not fully meet PLOS ONE’s publication criteria as it currently stands. Therefore, we invite you to submit a revised version of the manuscript that addresses the points raised during the review process.

We look forward to receiving your revised manuscript.

Kind regards,

Latika Gupta

Academic Editor

PLOS ONE

Additional Editor Comments :

Please detail survey pilot testing and validation using standard reporting guidelines. Rest of the revision seems adequate.

---

## [Author Response · Author response to Decision Letter 1]

25 Jul 2022

Editor Comments:

Please detail survey pilot testing and validation using standard reporting guidelines. Rest of the revision seems adequate

Author responses: 

COVID-19-related information consists of twelve questions. We run a pilot study for these 12 questions before rolling it out for this study. Before the initiation of data collection, we conducted a pilot study to determine whether the study's questionnaires were understandable for the general public, particularly for university students, and to examine the viability of the data collection tools for gathering data swiftly without imposing onerous conditions. Upon completing the pilot study, we revised some questions and altered wordings as per the convenience of the respondents. In addition, the pilot study was also conducted to determine whether the data showed too much or too little variability and to cross-check the eligibility criteria for the respondents. To assure validity, the researchers committed to upholding the study's credibility at every stage of the data collection and analysis process. We further reinforced the credibility by ensuring that each respondent understood the questionnaire and the purpose of the study. Both reliability and sensitivity were good. The Cronbach alpha for these questions was 0.864.

---

## [Editor Report · Decision Letter 2]

1 Aug 2022

Factors Associated with Facebook Addiction among University Students amid the COVID-19 Pandemic: Findings from an Online Cross-Sectional Survey

PONE-D-22-08770R2

Dear Dr. Ripon

We’re pleased to inform you that your manuscript has been judged scientifically suitable for publication and will be formally accepted for publication once it meets all outstanding technical requirements.

Kind regards,

Dr Latika Gupta

Academic Editor

PLOS ONE
---

## [Editor Report · Acceptance letter]

16 Aug 2022

PONE-D-22-08770R2 

Factors Associated with Facebook Addiction among University Students amid the COVID-19 Pandemic: Findings from an Online Cross-Sectional Survey 

Dear Dr. Ripon:

I'm pleased to inform you that your manuscript has been deemed suitable for publication in PLOS ONE. Congratulations! Your manuscript is now with our production department. 

Kind regards, 

on behalf of

Dr. Latika Gupta 

Academic Editor

PLOS ONE